ecology, ecosystems, environmental science

*Salmo salar*, bivariate time series, temporal trends, climate change, phenology, run conditions

**Author for correspondence:**
Elorri Arevalo
e-mail: elorri.arevalo@gmail.com

# Does global change increase the risk of maladaptation of Atlantic salmon migration through joint modifications of river temperature and discharge?

Elorri Arevalo[1], Anthony Maire[2], Stéphane Tétard[3], Etienne Prévost[4], Frédéric Lange[4], Frédéric Marchand[5], Quentin Josset[6,7,8] and Hilaire Drouineau[1]

[1]INRAE, Unité EABX–Écosystèmes Aquatiques et Changements Globaux, HYNES (Irstea-EDF R&D), 50 avenue de Verdun, 33612 Cestas Cedex, France
[2]EDF Recherche et Développement, Laboratoire National d'Hydraulique et Environnement, HYNES (Irstea-EDF R&D), 6 quai Watier, 78401 Chatou Cedex, France
[3]ICEO Environnement, 220 rue des Ailes, 85440 Talmont-Saint-Hilaire, France
[4]Université de Pau et des Pays de l'Adour, e2s UPPA, INRAE, ECOBIOP, Saint-Pée-sur-Nivelle, France
[5]INRAE, Unité Expérimentale d'Écologie et d'Écotoxicologie Aquatique, 65, rue de Saint-Brieuc, 35042 Rennes CEDEX, France
[6]UMR BOREA 7208, Muséum National D'Histoire Naturelle, Service des Stations Marines, 35800 Dinard, France
[7]MIAME — Management of Diadromous Fish in their Environment, OFB, INRAE, Institut Agro, UNIV PAU & PAYS ADOUR/E2S UPPA, Rennes, France
[8]Office Français de la Biodiversité, Direction Recherche et Appui Scientifique, Rue des Fontaines, 76260 Eu, France

EA, 0000-0002-7344-9330; AM, 0000-0003-0920-773X

In freshwater ecosystems, water temperature and discharge are two intrinsically associated triggers of key events in the life cycle of aquatic organisms such as the migration of diadromous fishes. However, global changes have already profoundly altered the thermal and hydrological regimes of rivers, affecting the timing of fish migration as well as the environmental conditions under which it occurs. In this study, we focused on Atlantic salmon (*Salmo salar*), an iconic diadromous species whose individuals migrate between marine nursery areas and continental spawning grounds. An innovative multivariate method was developed to analyse long-term datasets of daily water temperature, discharge and both salmon juvenile downstream and adult upstream migrations in three French rivers (the Bresle, Oir and Nivelle rivers). While all three rivers have gradually warmed over the last 35 years, changes in discharge have been very heterogeneous. Juveniles more frequently used warmer temperatures to migrate. Adults migrating a few weeks before spawning more frequently used warm temperatures associated with high discharges. This has already led to modifications in preferential niches of both life stages and suggests a potential mismatch between these populations' ecological preference and changes in their local environment due to global change.

## 1. Introduction

Most life cycle events of species are synchronized by abiotic factors, most of which are affected by global change [1–3]. In addition to photoperiod, water temperature and discharge are key triggers in riverine systems for the migration of organisms between reproductive, feeding or seasonal refuge habitats [4–6]. Water temperature governs the physiology and the internal state of ectothermic organisms (e.g. metabolism, energy status, hormonal control, stress level; [7,8]),

which trigger sexual maturation and migration [9,10]. Water temperature also affects the energy cost of migration and the swimming performance of organisms [11,12], while river discharge and water velocity stimulate migration and condition the ability of individuals to move and disperse [13]. Thus, natural selection has promoted the evolution of environmental preferenda leading to species migrating under conditions of water temperature and discharge that favour their survival. The interaction between global climate change and local anthropogenic pressures (including water withdrawal for human consumption, irrigation, industry and hydropower production) has altered thermal and hydrological regimes of many rivers throughout the world. This may have altered the occurrence of water temperature and discharge associations favourable to the migration of aquatic organisms and resulted in changes in migration timing (phenological shift), adaptation to new environmental conditions (local adaptation) or species extirpation [5,14,15].

The Atlantic salmon (*Salmo salar*) is an emblematic anadromous fish species that spends part of its life in freshwaters environments for reproduction and juvenile rearing and the other part in marine environments for sub-adult growth before sexual maturation [14]. This strategy is thought to optimize the growth and survival of individuals by making use of the most appropriate habitats at each life stage [16]. After typically 1–4 years in freshwater [17], juveniles undergo morphological, physiological and behavioural changes, called smoltification, which are a pre-adaptation to the marine environment [18]. Once these changes are completed, the downstream migration of smolts occurs over a 3–10 week period between March and July [19,20]. Photoperiod determines the broad window for the transformation of smolts, then the synergistic effect between water temperature and photoperiod above a minimum temperature threshold triggers migration [6]. Intermediate discharges are favourable for migration and peak discharges at the beginning of the migration season can strongly drive smolts' migration [21–23]. After one or more years of growth at sea, spawners migrate upstream to their native river between February and October [24], when water temperature is between 5 and 25°C [25,26] and discharge varies [17]. This 'homing' behaviour is strong in Atlantic salmon [27] and favours reproductive isolation [28], i.e. populations are genetically different and adapted to local environmental conditions [29]. As such, populations can display heterogeneous responses to modifications in their environment, including those resulting from global change [5].

Recent studies have documented earlier spring migrations (i.e. downstream migration of smolts and upstream migration of spawners) related to warming [14,30–32]. Earlier spring migration can be concomitant with a higher risk of extreme hydrological events (i.e. large floods in spring [33]), resulting in additional energy costs [34]. Additionally, spawners migrating in late summer and autumn are likely to face warmer water temperatures combined with very low discharges, which can delay migration or force them to migrate when water temperatures are close to their thermal tolerance limit [26,35]. Together, these changes may lead to a reduction in individual fitness, population productivity and ultimately threaten the species' persistence [36,37].

Ecological niches encompass the environmental conditions favourable to a species' persistence or to the achievement of a key life cycle event [38]. The environment offers a wide range of conditions that we define as the 'available niche'. These environmental conditions could be preferentially selected or, conversely, rejected by the species. We identify selected conditions as the 'preferential niche'. The available niche can be seen as the environmental supply and the preferential niche as the demand of the species. To allow the optimal survival of a species, it is essential that the supply and the demand (i.e. the available and preferential niches) match, at least partially. The interaction of the available and preferential niches corresponds to the environmental conditions actually used by the species and could be identified as the 'effective niche' (also referred as 'realized niche'). Objectively quantifying the consequences of joint temporal trends in environmental variables on key life cycle events is challenging [39]. We proposed the 'Choc' method to reveal changes in joint environmental conditions within different ecological niches [15,40]. This original method was tested on the spawning migration of the European eel in two rivers from Northern Europe and effectively highlighted the growing mismatch between the conditions preferentially selected by eels to migrate and those actually used. In the present article, the 'Choc' method is improved to compare the environmental trends within ecological niches among several rivers. Here it is applied on two life cycle events of Atlantic salmon with distinct ecological requirements.

More specifically, we successively investigate (i) changes in the 'available niche' (i.e. the occurrence of water temperature and discharge associations during the migration periods), (ii) changes in the 'effective niche' (i.e. the conditions of water temperature and discharge associations indeed used by salmon to migrate) and (iii) discuss the implications of potential niche shifts relative to the 'preferential niche' (i.e. water temperature and discharge associations preferentially selected by salmon to migrate). The method was applied to three French coastal rivers (the Bresle, Oir and Nivelle rivers) where water temperature, discharge and salmon migration activity have been monitored daily since the mid-1980s. This allowed the comparison of three distinct populations to detect possible local adaptation and to highlight a potential vulnerability due to a mismatch between the ecological preferences of a population and its local environment.

## 2. Material and methods

### (a) Description of study sites

Daily water temperature, discharge and salmon passages recorded since the mid-1980s at three small coastal rivers (the Bresle, Oir and Nivelle rivers) located along with the French Atlantic coast were considered (figure 1). These three rivers are part of a long-term monitoring network, the French Environmental Research Observatory for Small Coastal Streams (ORE DiaPFC; https://www6.inrae.fr/diapfc/). The Bresle watershed covers an area of 748 km². The watershed is moderately urbanized (6.2%; table 1, data extracted from CORINE Land Cover [41]), largely dominated by agriculture (74.7%) and natural areas (19.1%). The Oir watershed covers an area of 87 km². The watershed is slightly urbanized (2.2%) with few natural areas (3.1%) and essentially dominated by agriculture (94.3%). The Nivelle watershed covers an area of 238 km². The watershed is moderately urbanized (7.3%) or used for agriculture (34.1%) and essentially composed of natural areas (58.6%). The hydrology of the three rivers is largely influenced by rainfall, with a gradual decrease in discharge from January to June, followed by a period of low discharge in summer and an increase in autumn (see the mean monthly

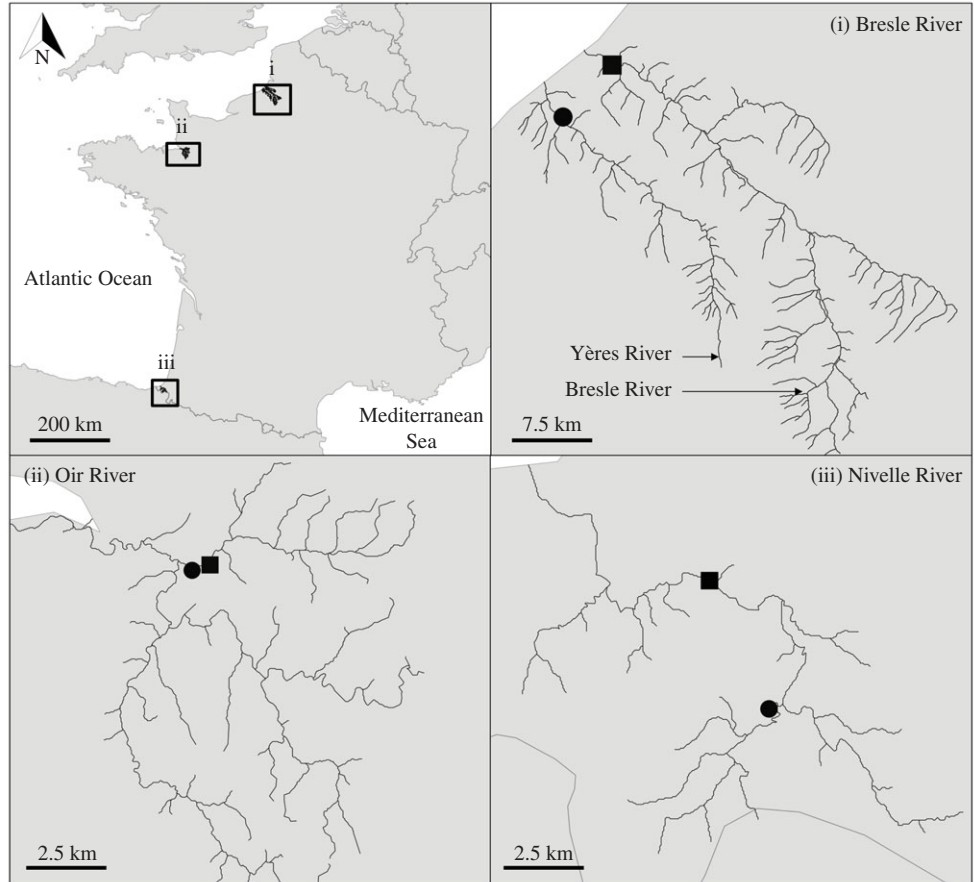

**Figure 1.** Location of discharge (circle) and water temperature (square) stations on the Bresle, Oir and Nivelle rivers. Fish traps are located at the same location as the water temperature stations.

discharge of each river in the electronic supplementary material, figure S1). However, the Bresle River is located above a vast chalk aquifer which tends to regulate the discharge, notably by supplying water to the river in summer. River and watershed characteristics are provided in table 1.

## (b) Water temperature and discharge data

Water temperature was recorded at the fish monitoring stations operated by the ORE DiaPFC, from which daily averages were calculated for each station. Daily averages of river discharge were extracted from BanqueHYDRO (http://www.hydro.eau france. fr/) at the station closest to the fish monitoring stations. For the Bresle River, the discharge from the adjacent Yères River was used because the record was longer and the hydrology and geology of the two rivers are similar (discharges were checked for consistency across the periods when data were available for both rivers). When water temperature or discharge data were missing for more than three consecutive weeks, the corresponding year was excluded from the analyses. The data considered in the present study cover the periods 1985–2019, 1987–2019 and 1986–2019 for the Bresle, Oir and Nivelle rivers, respectively. A complete description of the data, including the number of years excluded from the analyses and the annual and seasonal averages of water temperature and discharge, is provided for each river in table 1.

## (c) Salmon passage data

A comprehensive history of these populations (stocking and anthropic pressures) is available in the electronic supplementary material, appendix A. On all three rivers, fish monitoring stations were equipped with fish traps which allow exhaustive monitoring of salmonid populations [42] and they are checked at least once a day. The Bresle and Oir rivers are equipped with a

double trapping facility, one for spawners migrating upstream and the other for smolts migrating downstream. The Nivelle River is equipped with a single trap for catching spawners migrating upstream. Therefore, no data were available for smolts migrating downstream on this river. Over the covered periods, an annual average of 1127 (s.d. = 1051) and 779 (s.d. = 469) smolts were caught on the Bresle and Oir rivers and 99 (s.d. = 64), 85 (s.d. = 73) and 132 (s.d. = 78) spawners on the Bresle, Oir and Nivelle rivers, respectively. To test for temporal trends in the annual number of smolts and spawners caught, Mann–Kendall trend tests were performed. In addition, to investigate trends in the timing of the upstream and downstream migration of salmon, Mann–Kendall trend tests were performed on the number of days from 1 January to the date of passage of 5, 50 and 95% of the total number of smolts and spawners caught each year (electronic supplementary material, figure S2).

## (d) Description of the Choc method

Changes in the occurrence of water temperature and discharge associations were quantified using the Choc method [40], which identifies the water temperature and discharge associations that have become significantly more or less frequent over time. To facilitate the comparisons among rivers and seasons, we transformed these time series using their empirical cumulative distribution functions (i.e. a value of 0.75 indicates a temperature or a discharge that was exceeded only 25% of the time) instead of analysing raw water temperature and discharge values (see [15] for such an analysis). By doing so, temperature and discharge are standardized on common scales.

The Choc method involves the successive use of kernel density estimates and Mann–Kendall trend tests (see [40] for a full description of the method). Briefly, kernel density estimates are used to quantify the frequency of occurrence of any associations of

**Table 1.** Description of the studied rivers, their watershed (drainage area, geology and land cover) and the data analysed. This table provides the time period covered, the number of years excluded and included in the analyses, the annual and seasonal averages of water temperature and discharge (mean ± s.d.) and the total and annual average number of smolts migrating downstream and spawners migrating upstream. Concerning the discharge of the Bresle River, the discharge from the adjacent Yères River was used.

| river | Bresle | Oir | Nivelle |
|---|---|---|---|
| location | France Upper Normandy | France Lower Normandy | France Basque country |
| watershed area (km$^2$) | 748 | 87 | 238 |
| river length (km) | 72 | 20 | 39 |
| predominant geology | chalk | schist and granite | schist and sandstone |
| land cover (%) | | | |
| urbanization | 6.2 | 2.2 | 7.3 |
| intensive agriculture | 52.5 | 52.8 | 10.1 |
| low-impact agriculture | 22.2 | 41.5 | 24.0 |
| natural areas | 19.1 | 3.5 | 58.6 |
| time period | 1985–2019 | 1987–2019 | 1986–2019 |
| number of years | | | |
| excluded | 0 | 3 | 1 |
| analysed | 35 | 30 | 33 |
| water temperature (°C) | | | |
| yearly | 11.7 ± 4.2 | 11.9 ± 3.6 | 13.9 ± 4.1 |
| winter | 7.5 ± 2.3 | 8.6 ± 1.9 | 9.9 ± 2.1 |
| spring | 13.7 ± 2.7 | 13.1 ± 2.3 | 15.0 ± 2.6 |
| summer | 16.2 ± 1.9 | 15.9 ± 1.8 | 18.7 ± 2.0 |
| autumn | 9.4 ± 2.8 | 10.0 ± 2.8 | 12.0 ± 2.9 |
| discharge (m$^3$ s$^{-1}$) | | | |
| yearly | 2.8 ± 1.5 | 1.1 ± 1.0 | 4.7 ± 6.6 |
| winter | 3.7 ± 1.8 | 1.9 ± 1.2 | 6.9 ± 8.0 |
| spring | 3.1 ± 1.4 | 0.9 ± 0.6 | 4.6 ± 5.5 |
| summer | 2.0 ± 0.6 | 0.4 ± 0.3 | 1.9 ± 2.8 |
| autumn | 2.3 ± 1.1 | 1.0 ± 1.1 | 5.4 ± 7.5 |
| smolts migrating downstream | | | |
| total number | 36 088 | 23 365 | — |
| annual average | 1127 | 779 | — |
| spawners migrating upstream | | | |
| total number | 3474 | 2561 | 4366 |
| annual average | 99 | 85 | 132 |

water temperature and discharge for each year. Then Mann–Kendall tests are used to detect temporal trends in these frequencies of occurrence. Finally, bootstraps are used to check the significance of the trends. This approach was applied independently on the three rivers and on the two salmon stages. Results at the annual scale (i.e. not restricted to migration periods) are presented in electronic supplementary material, appendix B.

### (e) Trends in water temperature and discharge associations during salmon migration periods: the available niche

To detect changes in the available niche, i.e. the occurrence of water temperature × discharge associations during the migration period, the Choc method was applied to daily

water temperature and discharge. Kernels estimated from the available niche were denoted Kde$_{avail}$. According to the date of passage of 5% and 95% of the total number of smolts caught, the downstream migration period was defined between March and May for all rivers (figure 2*b*; electronic supplementary material, figure S2). Similarly, the spawning migration period was defined between June and November on the Bresle River, between October and December on the Oir River and between April and November on the Nivelle River (figure 2*d*; electronic supplementary material, figure S2). Difference of spawning migration periods among rivers is related to the age structure of the populations: in the Oir River, most spawners have spent one winter at sea (which preferentially migrate in autumn), while there is a mix of fish that have spent one versus several winters at sea (which migrate preferentially in spring) in the two other rivers.

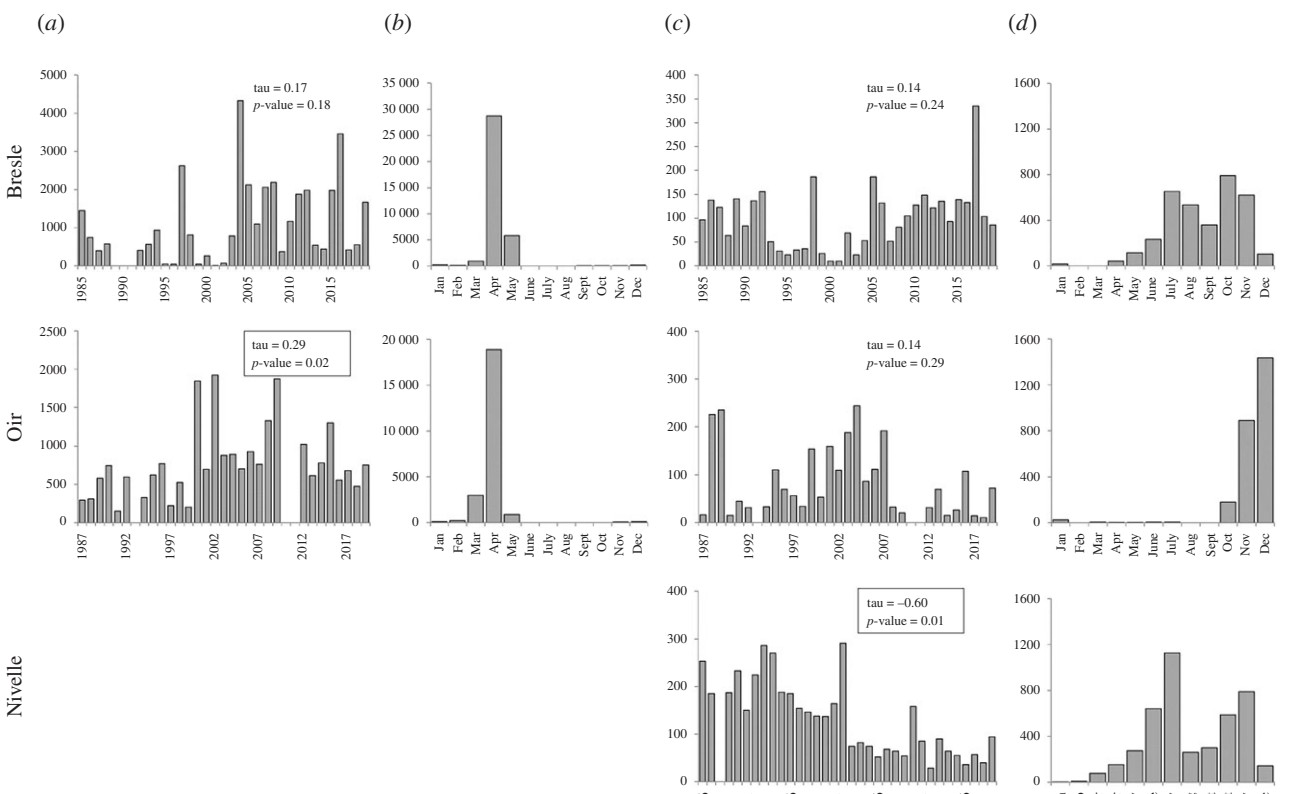

**Figure 2.** Description of Atlantic salmon populations migrating on the Bresle, Oir and Nivelle rivers: (*a*) annual number of smolts caught in the traps, (*b*) total smolt catch per month over the study period, (*c*) annual number of spawners caught in the traps and (*d*) total spawner catch per month over the study period. Mann–Kendall tau and *p*-value are indicated, with significant values boxed.

## (f) Trends in water temperature and discharge actually used by salmon to migrate: the effective niche

The daily proportions of smolts migrating downstream and spawners migrating upstream were calculated per year as the daily count divided by the total counts of the year. To detect changes in the environmental conditions used for migration, i.e. the effective niche, we carried out a second Choc analysis in which daily water temperature × discharge associations were weighted by their corresponding daily proportions of migrating salmon during the kernel estimation stage. Resulting kernels were denoted Kde$_{salmon}$.

## (g) Determination of the preferential niche for salmon migration and its potential changes over time

To characterize the preferential niche for smolts and spawners (i.e. water temperature and discharge associations preferentially selected by salmon to migrate regardless of the frequency of conditions actually observed), we built an electivity index based on the Ivlev feeding electivity index [43]. The electivity of a water temperature × discharge association $E'(T,Q)$ was calculated as follows:

$$E'(T,Q) = \frac{(\text{Kde}_{salmon}(T,Q) - \text{Kde}_{avail}(T,Q))}{(\text{Kde}_{salmon}(T,Q) + \text{Kde}_{avail}(T,Q))},$$

With Kde$_{avail}(T,Q)$ and Kde$_{salmon}(T,Q)$, the estimated kernel densities of probability of occurrence of water temperature $T$ and discharge $Q$ estimated based on available and effective niches. Index values close to −1 correspond to associations 'rejected' by salmon whereas values close to 1 indicate 'selected' associations. Index value was unreliable for low kernel estimations, so we ignored the 5% of the rarest associations in the available niche [15].

To get an overall picture of the preferential niche over the whole study period, electivity indices were first computed using kernel density estimations fitted on all years pooled together by salmon stages. Then, to detect temporal trends in the preferential niche, we computed yearly electivity indices by comparing Kde$_{salmon}$ and Kde$_{avail}$ of the corresponding year. This means that for a dataset covering 35 years, each environmental association had 35 electivity indices corresponding to each year of the dataset. Then, Mann–Kendall trend tests were applied to the time series of the electivity indices of each association and statistical significances were assessed through a random permutation procedure. The entire methodology is implemented in the R software [44] 'chocR' package [45]. Percentages of significant changes in the occurrence of water temperature × discharge associations in the three niches are provided in the electronic supplementary material, table S1.

## 3. Results

### (a) Spring smolt migration

Since the 1980s, the number of smolts caught remained relatively stable on the Bresle River and increased significantly on the Oir River (Mann–Kendall trend test *p*-value = 0.18 and 0.02, respectively; figure 2*a*). The smolt migration period (between March and May for both rivers) remained stable over time on the Oir river (*p*-values not significant; electronic supplementary material, figure S2), whereas through time it progressively ended later on the Bresle River (*p*-value of the passage dates of 95% of the population = 0.01).

The available niche during smolt migration on the Bresle River has changed little over the study period (figure 3), except for a rarefaction of cold temperatures associated with low discharges. This change was not detected in the effective niche, in which we observed that migrations occurring at high temperatures have become significantly more frequent.

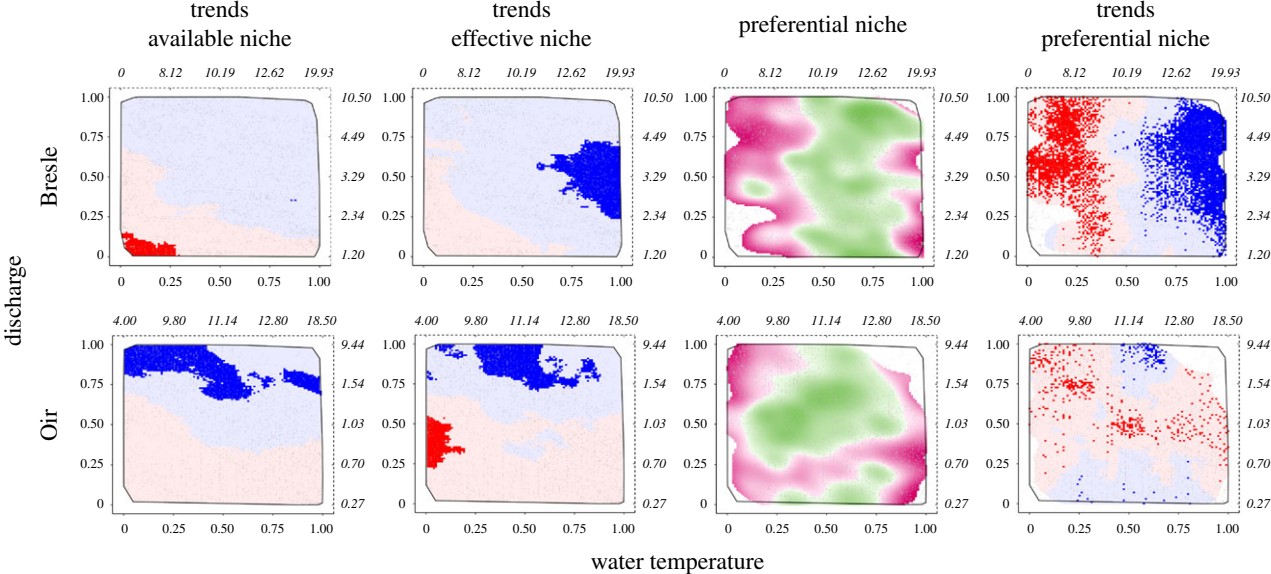

**Figure 3.** Conditions of downstream migration of smolts of Atlantic salmon on the Bresle and Oir rivers. These two-dimensional heat maps of water temperature (X-axis) and discharge (Y-axis) were computed for the downstream migration period (i.e. from March to May) and represent the trends in the available niche, the trends in the effective niche, the average preferential niche and the trends in the preferential niche. The scale on the X and Y axes represents the quantiles of the empirical cumulative distribution for water temperature and discharge. Values in natural scale are shown in italics (in °C and $m^3 \, s^{-1}$, respectively) on the dashed third and fourth axes. Discharge and temperature associations that have become more or less frequent over the study period are shown in blue or red, respectively. Light and dark colours correspond to non-significant and significant trends, respectively (e.g. dark red corresponds to significant decreasing trends, light blue to non-significant increasing trends). The grey line delineates the convex hull (i.e. the smallest space encompassing all the points of the dataset). For the preferential niche, the 5% less frequent associations are represented in white colour inside the convex hull. Selected and rejected associations are shown in green or purple, respectively. (Online version in colour.)

According to the preferential niche, smolts preferentially selected water temperatures above 9°C to migrate. Warm temperatures, which were more frequently used by smolts, gradually led to a significant change in the preferential niche: smolts more preferentially selected warm temperatures more recently than at the beginning of the study period. On the Oir River, changes in the available niche directly influenced the effective niche, with the highest discharges occurring more frequently and being more frequently used by smolts to migrate. As on the Bresle River, smolts preferentially selected water temperatures above 9°C to migrate. Unlike on the Bresle River, no clear trend was observed in the preferential niche.

## (b) Spring and autumn adult migrations

Since the 1980s, the number of spawners caught has remained stable on the Bresle and Oir rivers, while it has decreased significantly on the Nivelle River (Mann–Kendall trend test $p$-value = 0.24, 0.29 and 0.01, respectively; figure 2c). As mentioned earlier, the migration seasons are distinct among rivers: from June and November on the Bresle River, between October and December on the Oir River and between April and November on the Nivelle River. The spawning migration periods did not change significantly for any rivers over the study period (all $p$-values of the passage dates of 50% of the population greater than 0.05; electronic supplementary material, figure S2).

On the Bresle River, there was little change in the frequency of associations in both the available and effective niches (figure 4). Spawners preferentially selected cold temperatures (below 14°C) regardless of the discharge, and warm temperatures (above 16°C) associated with intermediate/high discharge. The preferential niche has also changed little. On the Oir River, intermediate and high discharges have become significantly more frequent in the available niche, which had

direct consequences on the effective niche. Spawners preferentially selected intermediate and high discharges and rejected low discharges. Changes in the preferential niche confirmed that low discharges were increasingly rejected by spawners and warm temperatures were increasingly selected. On the Nivelle River, high discharge associated with cold temperature has become significantly scarcer, while low discharge associated with intermediate and high temperatures has become more frequent. These changes did not affect the effective niche and we observed that warm temperatures associated with the lowest discharge were less frequently used by spawners. Spawners preferentially selected intermediate and high discharges and rejected low discharges, regardless of the water temperature. The preferential niche of spawners has changed substantially: spawners have more frequently selected cold temperatures associated with high discharge, while they have increasingly rejected various water temperature × discharge associations on the margins of the preferential niche (figure 4).

## 4. Discussion

### (a) Long-term changes in hydrological and thermal regimes

France is at the southern edge of the geographic range of Atlantic salmon [46]. A gradual rise in air temperature has been observed over the last decades [47], significantly increasing the water temperature of rivers by up to 1°C per decade [48,49]. This has been observed on the Bresle, Oir and Nivelle rivers, where the coldest water temperatures have become scarcer and the warmest more frequent since the 1980s (see electronic supplementary material, appendix B for a complete presentation of the environmental trends at the annual scale). The effect of global changes (e.g. changes

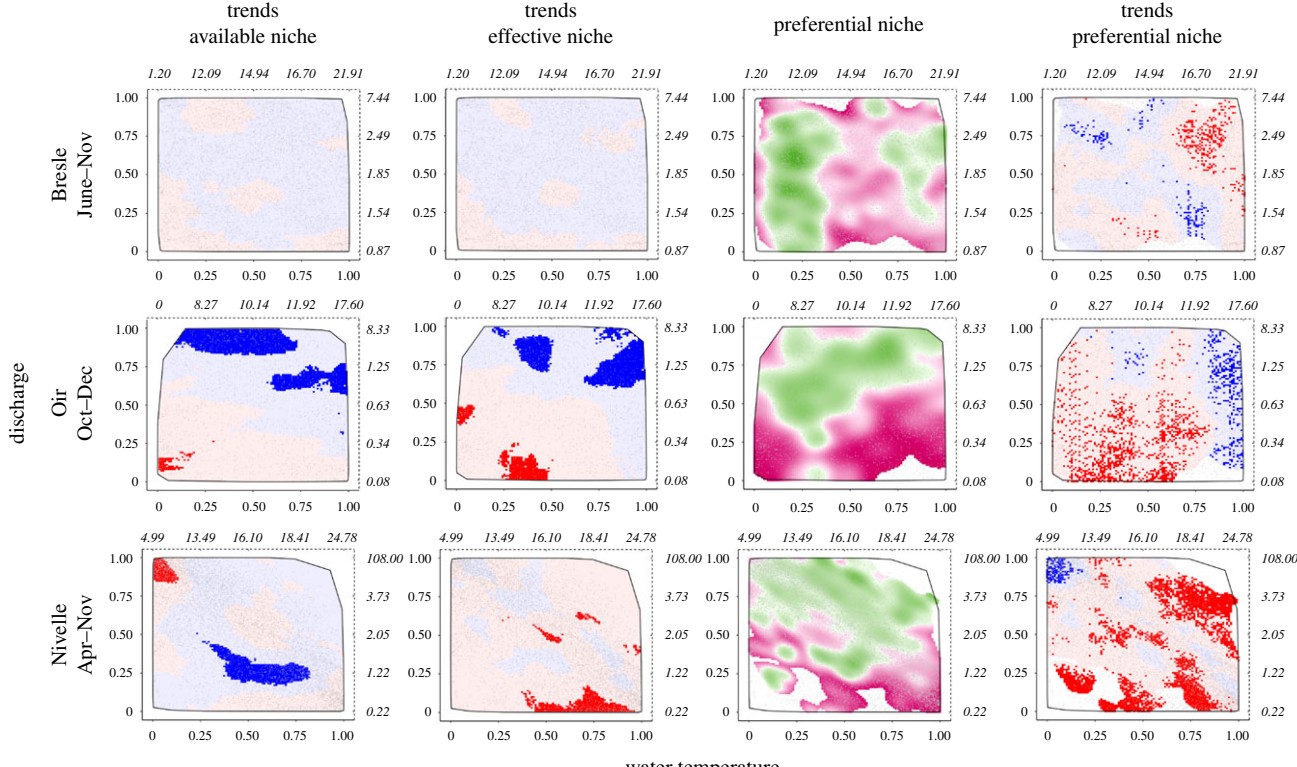

**Figure 4.** Conditions of spawning migration of adults of Atlantic salmon on the Bresle, Oir and Nivelle rivers. These two-dimensional heat maps of water temperature (X-axis) and discharge (Y-axis) were computed for the upstream migration season and represent the trends in the available niche, the trends in the effective niche, the average preferential niche and the trends in the preferential niche. The scale on the X and Y axes represents the quantiles of the empirical cumulative distribution for water temperature and discharge. Values in natural scale are shown in italics (in °C and $m^3 s^{-1}$ respectively) on the dashed third and fourth axes. Discharge and temperature associations that have become more or less frequent over the study period are shown in blue or red, respectively. Light and dark colours correspond to non-significant and significant trends, respectively. The grey line delineates the convex hull. For the preferential niche, the 5% less frequent associations are represented in white inside the convex hull. Selected and rejected associations are shown in green or purple, respectively. (Online version in colour.)

in thermal regime of rivers or in the precipitation levels) on hydrological regimes are heterogeneous and depend on the region and other local anthropogenic pressures [50]. Few significant hydrological changes have been observed in the Bresle River because it is located above a vast chalk aquifer, which gives it a unique flow regime [51]. The aquifer naturally buffers the discharge by smoothing out short-term climatic variations such as heavy rainfall [52] and the aquifer also regulates the water temperature of the river [53]. High discharge from the Oir River, including extreme floods, has become more frequent, while the Nivelle River has been subjected to more frequent drought situations (i.e. low discharge × warm temperature; electronic supplementary material, appendix B and see 'Trends in available niche' figures 3 and 4). It is now clearly established that global changes are leading to more intense summer droughts that extend later in the autumn throughout southern Europe [54–56]. The multiplication of these extreme events, including floods, droughts and heatwaves, profoundly affects freshwater ecosystems and all the organisms that depend on them to complete their life cycle, including migratory fish species [55,57–59].

## (b) Deleterious changes in the conditions of downstream migration of smolts

We highlighted that water temperature and discharge conditions during which smolts of Atlantic salmon preferentially migrate to the sea are similar among the rivers studied, i.e. water temperature between 9 and 15°C and high discharge. These preferentially selected conditions are

consistent with existing observations [20,31]. Many studies have revealed that the downstream migration of smolts seems synchronized with similar environmental cues among populations probably to allow them to reach the sea when sea water temperatures are above 8°C, thus improving feeding conditions, salinity tolerance and swimming performance [60–62]. We considered only two environmental factors, water temperature and discharge, which are among the most important in triggering salmon migration [63]. Other factors can drive the motivation to migrate; for example Teichert et al. [6] considered photoperiod and water temperature simultaneously and they projected an earlier smolt migration and an extension of the migration period due to climate change. Photoperiod is correlated with water temperature and allows individuals to synchronize to season. Therefore, we have integrated it indirectly by focusing only on the spring migration period.

An extension of the spring migration season as observed on the Bresle River (electronic supplementary material, figure S2; quantile 95%) increases the risk of migrating at warmer water temperatures (i.e. more summer-like temperatures), leading to higher energy costs. Conversely, an earlier migration as observed on the Oir River (electronic supplementary material, figure S2; quantiles 5% and 95%) could result in smolts migrating at smaller sizes, making them potentially more vulnerable and less mobile [14]. In both cases, the smaller length and poorer physical condition of smolts reaching the sea (as shown by the decreasing trends in smolt fork length from 1985 to 2019 on the Bresle and Oir rivers; electronic supplementary material, figure S3) could impair their ability to swim and escape predators, therefore impacting their survival rate [64]

and the resilience of smolts to potentially inhospitable marine conditions [65]. Moreover, the influence of global change on river systems and marine environments could lead to a growing mismatch between downstream migration triggers and suitable marine environmental conditions, with potential implications for adult return rates [66] and individual fitness [67].

## (c) Diversity of spawning migration strategies and evolutionary potential

The diversity of migration strategies is reflected in part by the age structure of the population. Spawners having spent one winter at sea (1SW) and those having spent multiple winters at sea (MSW) have specific migration periods and conditions. However, only 13% of the total number of spawners are MSW salmon on the Oir River and only 35% of the spawners caught on the Bresle River have been aged, which prevented us from giving too much emphasis on these results (electronic supplementary material, figure S4). The heterogeneous environmental conditions observed on the three rivers studied may exert divergent selective pressures on the age structure of their respective salmon population [68,69]. This divergence is highlighted through both a migration period and a preferential niche specific to each river. The timing of the spawning migration is a critical life-history trait with a strong genetic basis [70] and the evolutionary potential of a population depends largely on the duration of the migration season [71]. When spawners migrate during several months (both spring and autumn running spawners), they have more possibility to adapt their timing to match favourable environmental conditions. This likely explains the limited number of associations more frequently used in the effective niche on the Bresle and Nivelle rivers. By contrast, when migration is limited to a short period, as observed on the Oir River, the hormonal stimulation of adults probably does not allow them to delay their migration [72] and they thus attempt to reach the spawning grounds regardless of the suitability of environmental conditions. Moreover, in such rivers where the migration season is close to the spawning period, delaying the migration to wait for suitable environmental conditions or the occurrence of a catastrophic event would threaten the presence of adults on the spawning grounds in time and ultimately compromise the recruitment of the following year. This is probably why we have observed an increasing use of warmer water temperature by spawners on the Oir River, which may lead to increased energy costs and pre-spawning mortality. Additionally, a short migration period is generally associated with low genetic variability [71] and poor adaptive capacity to a stochastic environment [73,74]. The situation for the salmon populations of the Nivelle and Bresle rivers is likely to be less alarming as long as the diversity of migration strategies (with salmon migrating in spring and autumn) is maintained. However, the number of spawners is constantly decreasing on the Nivelle River, especially since the 2000s [75]. This does not yet affect the number of juveniles observed in autumn, but this decrease may threaten the diversity of migration strategies within the population. Phenological diversity within and between populations contributes to portfolio effects that buffer salmon abundances from environmental variability [71,76,77]. However, ongoing global changes involve rapid environmental modifications, with the main concern being that these changes may be too abrupt to allow salmon populations to adapt. This is even more critical in populations of small systems such as those in this study, where natural diversity is likely to be limited, and has been further reduced by all anthropogenic pressures.

## 5. Conclusion

Diadromous fish populations such as Atlantic salmon are in drastic decline worldwide due to the effects of global change on both freshwater and marine environments [78,79]. Changes in one life stage can propagate throughout the life cycle and affect subsequent life stages or accumulate over several generations [80]. Our analyses, using the Choc method, highlighted that changes in environmental conditions have particularly affected the migration events of smolts on the Bresle River and of spawners on the Oir River. In both cases, the evolution of the preferential niche towards warmer temperatures suggests that these vulnerable populations require special attention. Applied to more anthropized systems, our method can help implement management measures to maintain optimal discharges during migratory events. The declines in size of these populations are alarming and it remains crucial to further characterize the pressures exerted on them.

Data accessibility. Data are freely available at the following links: Bresle and Oir rivers: https://doi.org/10.15454/1.5573930653786494E12 and Nivelle River: https://doi.org/10.15454/1.5572402068944548E12.

Authors' contributions. E.A.: conceptualization, data curation, formal analysis, methodology, validation, visualization, writing the original draft, writing the review and editing; A.M.: conceptualization, funding acquisition, methodology, project administration, supervision, validation, visualization, writing – original draft, writing the review and editing; S.T.: conceptualization, funding acquisition, methodology, project administration, supervision, validation, writing the review and editing; E.P.: validation, visualization, writing the review and editing; F.L.: validation, visualization, writing the review and editing; F.M.: validation, visualization, writing the review and editing; Q.J.: validation, visualization, writing the review and editing; H.D.: conceptualization, formal analysis, funding acquisition, methodology, project administration, software, supervision, validation, visualization, writing – original draft, writing the review and editing.

All authors gave final approval for publication and agreed to be held accountable for the work performed therein.

Competing interests. We declare we have no competing interests.

Funding. The present study was funded by the HYNES programme (INRAE – EDF R&D). We would like to thank Électricité de France (EDF) for their financial support.

Acknowledgements. All methods involved in the trapping and handling of salmon have been carried out in accordance with relevant guidelines and regulations (APAFIS#26834-2020080615584492 v3 for the Bresle and Oir rivers and APAFIS#8949-2017021711391209 v3 for the Nivelle River). All experimental protocols involved in the trapping and handling of salmon comply with the EU Animal Welfare Directive 2010/63/EU and the French Rural and Maritime Fishing Code, in particular Articles R.214-87 to R.214-126.

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
