## [Peer Review File · Proceedings of the Royal Society B: Biological Sciences]

Review History

RSPB-2021-1882.R0 (Original submission)

Review form: Reviewer 1

Recommendation

Accept with minor revision (please list in comments)

Scientific importance: Is the manuscript an original and important contribution to its field?

Good

General interest: Is the paper of sufficient general interest?

Good

Quality of the paper: Is the overall quality of the paper suitable?

Good

Is the length of the paper justified?

Yes

Should the paper be seen by a specialist statistical reviewer?

No

Do you have any concerns about statistical analyses in this paper? If so, please specify them explicitly in your report.

No

It is a condition of publication that authors make their supporting data, code and materials available - either as supplementary material or hosted in an external repository. Please rate, if applicable, the supporting data on the following criteria.

Is it accessible?

Yes

Is it clear?

Yes

Is it adequate?

Yes

Do you have any ethical concerns with this paper?

No

Comments to the Author

General comments:

This is a good study describing a method to assess the effects of anthropogenic change on aquatic organisms (Atlantic salmon). It should be of broad general interest to conservation-minded biologists.

In order to fully convey the relevance of the analyses undertaken, the authors need to be more explicit about the case studies in both the Methods and the Discussion:

A brief history and summary of population trends must be provided for each study river. Of particular relevance is 1) what were the historic population sizes and distributions within each watershed, and 2) a discussion of how the newly introduced populations (whether from other rivers or from hatcheries) likely differ from the original endemic populations in locally-adapted traits, particularly run timing and age at maturity.

Such small populations as found in the present study (100 adult spawners, probably below minimum viable population size) are likely far below historic levels when locally-adapted subpopulations were probably found throughout each watershed. Each of these subpopulations would have had unique migration adaptations (some of which would have led to a "portfolio effect" a la Schindler et al. 2010, against climate change).

So, while the Choc method may be suitable for understanding trends for the current populations, the authors should be explicit that historic populations would likely have been far more diverse and resilient to climate change (or other anthropogenic factors).

A few specific comments:

Line 33 "a" few weeks

Lines 41-43: add "in addition to photoperiod" and appropriate citations

Lines 48-49: not "emergence" - "evolution" is the correct word

Lines 50-52: "Global change (including climate change and other anthropogenic pressures such as water withdrawal for human consumption, irrigation, industry and hydropower production)"

Global change does not include anything that is not global. These other are regional or local!

Lines 64-73: migration timing is a heritable trait, mediated by biotic (internal state) and abiotic (env) factors. Local adaptation in migration timing should be clearly stated here (since it is these local adaptations that are subject to mismatch due to climate change). Again, the historic conditions in the case study watersheds will dictate to some extent the relevance of the analyses.

Lines 84- why not use the widely accepted terms functional and realized niche?

Lines 312-317; "In both cases, the smaller length and poorer physical condition of smolts reaching the sea could impair their ability to swim and escape predators, therefore impacting their survival rate [73] and the resilience of smolts to inhospitable marine conditions [74]. Moreover, global changes could lead to a growing mismatch between downstream migration triggers and suitable marine environmental conditions, with potential implications for adult return rates [75] and individual fitness [76,77]."

Is there any evidence compare the historic conditions of smolt size and status, and migration timing mismatch, to the present, highly-modified and small populations? That is, smolt size and timing of these new, small populations might differ substantially from historic populations, that have evolved over thousands of years.

Lines 318- "Diversity of spawning migration strategies and evolutionary potential" :

Again, a discussion of the diversity of the historic, endemic populations is warranted here. How much "shifting baselines" have you experienced. Are populations of 100 spawners even viable? How can the study help recommend restoration of truly resilient populations of 1000 or more spawners in these watersheds?

Lines 349- Conclusions:

Do you consider the 100 spawners (down from 300) returning to the Nivelles River as NOT ALARMING? You should be less timid in drawing inferences from your study in terms of management. I would consider all of these populations to be exhibiting rather alarming trends.

Review form: Reviewer 2

Recommendation

Accept with minor revision (please list in comments)

Scientific importance: Is the manuscript an original and important contribution to its field?

Excellent

General interest: Is the paper of sufficient general interest?

Good

Quality of the paper: Is the overall quality of the paper suitable?

Excellent

Is the length of the paper justified?

Yes

Should the paper be seen by a specialist statistical reviewer?

No

Do you have any concerns about statistical analyses in this paper? If so, please specify them explicitly in your report.

No

It is a condition of publication that authors make their supporting data, code and materials available - either as supplementary material or hosted in an external repository. Please rate, if applicable, the supporting data on the following criteria.

Is it accessible?

Yes

Is it clear?

Yes

Is it adequate?

Yes

Do you have any ethical concerns with this paper?

No

Comments to the Author

This clear and well written article uses a previously developed and deployed methodology (Choc) to examine the effects of systematic environmental variation (river temperature and discharge) due to primarily anthropogenic impacts upon the migration patterns of smolt and adult Atlantic salmon. The Choc method, in analyzing patterns based on how changes in the environment are affecting individual animal's environmental niches and the consequent impact of the changes upon population parameters, provides a predictor of population trends that is rooted in ecological (niche) theory. This offers a more satisfying and mechanistic approach to understanding the species' population dynamics compared to the more frequently used correlative approaches. Most of my comments are editorial in nature.

Regarding the title, It seems to me it is not the migration that is in jeopardy, but individual survival during the course of migration and it is survival probabilities that the authors have investigated. I suggest slightly changing the title to capture this, perhaps "Can global change jeopardize Atlantic salmon survival during migrations through joint modifications of river temperature and discharge?"

l. 25 add "fish" before "migration"

l. 33 add "a" before "few"

l. 35 strike the first "then" then add "both life stages" before "preferential". Does that capture what the authors are trying to state? Also, add "these" before "population's".

l. 36 change to "... changes in their local environments due to global change."

l. 54 change "associations" to "combinations" and "subsequently" to "consequently"?

Associations works for the rest of the MS

l. 56 add "The" before "Atlantic salmon"

l. 60 add "typically" before "1-4 years"

l. 62 change "is" to "are"

l. 65 add "minimum" before "temperatures"

l. 70 change "The" to "This"

l. 82 change "population's" to "population"

l. 83 change "their" to "the species' "

l. 88 change "this affinity of the species to the available" to "selected"

l. 101 change to "niches among several rivers. Here it is applied....."

l. 106 change "use" to "used"

- l. 145 Were the counting facilities enumerating all fish coming up the river? Or did they only cover part of the river and hence generating a partial count which may not scale among years predictably to the total population size?
- l. 153 How did you factor this variable stocking effort into your analysis and among-river comparisons? The numbers are small and perhaps were not expected to appreciably affect either smolt output or adult returns? Please explain.
- l. 161 change “salmons” to “salmon”
- l. 193 to read “ there is a mix of fish that have spent one versus several winters at sea....”
- l. 213 change “salmons” to “salmon”
- l. 229 add “relatively” before “stable”.
- l. 230 So the Oir River that had no stocking showed a significant increase in smolt production? What is your interpretation of this?
- l. 236 change “expect” to “except for”
- l. 284 change “original” to “unique”
- l. 290 Can you make this general statement of “profound effects” in the context of the work that you present? You have shown a slow increasing temperature trend and no general patterns for discharge. Are these profound changes?
- l. 299 change “between” to “among”
- l. 300 change “trophic availability” to “feeding conditions”
- l. 303 change to read “drive the motivation to migrate; for example Teichert et al...”
- l. 305 to read “migration and a prolongation of the migration period due to climate change....”
- l. 308 to read “A longer spring migration season such as that of the Bresle River.....”
- l. 314 add “potentially” before “inhospitable”
- l. 322 change “salmons” to “salmon”
- l. 323 strike “at the moment”
- l. 327 add “the” before “spawning”
- l. 328 - 348 I do not follow the logic of the statement that the evolutionary potential depends large on the duration of the migration season. Surely it depends on the genetics of the population? This needs to be better explained and defended. The argument surrounding the better capability of a population to adapt if the population has a longer migration period works depending on the definition that is being used for a salmon “population”. If particular tributaries of the system have animals that migrate at different genetically determined times, and conditions deteriorate in the river such that particular runs that occurred at times that are now unfavorable disappear and those occurring in more favorable times surge, is this actually a “population” adaptation as opposed to a replacement of populations? Perhaps the salmon populations in your river are not structured into distinct population segments?
- l. 336 change “river” to “rivers”
- Conclusion and l. 358 in particular. According to Table 1, the Nivelles River is by far the least disturbed system in which you worked, having 58.6% of its watershed in “natural areas” compared to 3.5% and 19.1% for the Oir and Bresle Rivers, respectively, yet the Nivelles has the highest annual temperatures which seems curious to me. Have these percentages changed over the duration of the study period, or has there been a loss of natural habitat during this time? Since stocking of hatchery fish and habitat restoration are probably the two major activities currently employed for Atlantic salmon restoration, can your methodology and results inform either of these practices?
- Table 1, last entry on “Spawners migrating upstream”, did the years included in the total number for each river system vary among systems?
- Fig 1 Bresle River panel. To avoid confusion please individually label the Bresle and Yères Rivers in this figure.
- Fig 2 and 4 caption. Are the values presented catches? Or are they the counts in the various counting systems used on the different rivers? I would rephrase the description of the axes on both Figures to read that the X and Y axes plot your standardized scores, and the italicized secondary axes plot true values of the variables. The term “natural” is confusing.
- Fig 4 caption. Specify that this Figure is for adult salmon.

Decision letter (RSPB-2021-1882.R0)

14-Oct-2021

Dear Dr Arevalo:

Your manuscript has now been peer reviewed and the reviews have been assessed by an Associate Editor. The reviewers' comments (not including confidential comments to the Editor) and the comments from the Associate Editor are included at the end of this email for your reference. As you will see, the reviewers and the Editors have raised some concerns with your manuscript and we would like to invite you to revise your manuscript to address them.

Research ethics:

Use of animals and field studies:

It is a condition of publication that you make available the data and research materials supporting the results in the article. Please see our Data Sharing Policies (<https://royalsociety.org/journals/authors/author-guidelines/#data>). Datasets should be deposited in an appropriate publicly available repository and details of the associated accession number, link or DOI to the datasets must be included in the Data Accessibility section of the

article (<https://royalsociety.org/journals/ethics-policies/data-sharing-mining/>). Reference(s) to datasets should also be included in the reference list of the article with DOIs (where available).

If you wish to submit your data to Dryad (<http://datadryad.org/>) and have not already done so you can submit your data via this link [http://datadryad.org/submit?journalID=RSPB&manu=\(Document not available\)](http://datadryad.org/submit?journalID=RSPB&manu=(Document%20not%20available)), which will take you to your unique entry in the Dryad repository.

Please submit a copy of your revised paper within three weeks. If we do not hear from you within this time your manuscript will be rejected. If you are unable to meet this deadline please let us know as soon as possible, as we may be able to grant a short extension.

Best wishes,
Dr Daniel Costa
mailto: proceedingsb@royalsociety.org

Associate Editor
Comments to Author:

Both reviewers consider this manuscript favourably. Both have provided useful comments for improving the manuscript that should be considered by the authors and incorporated into a revised manuscript.

Reviewer(s)¹ Comments to Author:
Referee: 1

Comments to the Author(s)
General comments:

This is a good study describing a method to assess the effects of anthropogenic change on aquatic organisms (Atlantic salmon). It should be of broad general interest to conservation-minded biologists.

In order to fully convey the relevance of the analyses undertaken, the authors need to be more explicit about the case studies in both the Methods and the Discussion:

A brief history and summary of population trends must be provided for each study river. Of particular relevance is 1) what were the historic population sizes and distributions within each watershed, and 2) a discussion of how the newly introduced populations (whether from other rivers or from hatcheries) likely differ from the original endemic populations in locally-adapted traits, particularly run timing and age at maturity.

Such small populations as found in the present study (100 adult spawners, probably below minimum viable population size) are likely far below historic levels when locally-adapted sub-populations were probably found throughout each watershed. Each of these subpopulations would have had unique migration adaptations (some of which would have led to a “portfolio effect” a la Schindler et al. 2010, against climate change).

So, while the Choc method may be suitable for understanding trends for the current populations, the authors should be explicit that historic populations would likely have been far more diverse and resilient to climate change (or other anthropogenic factors).

A few specific comments:

Line 33 “a” few weeks

Lines 41-43: add “in addition to photoperiod” and appropriate citations

Lines 48-49: not “emergence” – “evolution” is the correct word

Lines 50-52: “Global change (including climate change and other anthropogenic pressures such as water withdrawal for human consumption, irrigation, industry and hydropower production)”

Global change does not include anything that is not global. These other are regional or local!

Lines 64-73: migration timing is a heritable trait, mediated by biotic (internal state) and abiotic (env) factors. Local adaptation in migration timing should be clearly stated here (since it is these local adaptations that are subject to mismatch due to climate change). Again, the historic conditions in the case study watersheds will dictate to some extent the relevance of the analyses.

Lines 84- why not use the widely accepted terms functional and realized niche?

Lines 312-317; “In both cases, the smaller length and poorer physical condition of smolts reaching the sea could impair their ability to swim and escape predators, therefore impacting their survival rate [73] and the resilience of smolts to inhospitable marine conditions [74]. Moreover, global changes could lead to a growing mismatch between downstream migration triggers and suitable marine environmental conditions, with potential implications for adult return rates [75] and individual fitness [76,77].”

Is there any evidence compare the historic conditions of smolt size and status, and migration timing mismatch, to the present, highly-modified and small populations? That is, smolt size and timing of these new, small populations might differ substantially from historic populations, that have evolved over thousands of years.

Lines 318- “Diversity of spawning migration strategies and evolutionary potential” :

Again, a discussion of the diversity of the historic, endemic populations is warranted here. How much “shifting baselines” have you experienced. Are populations of 100 spawners even viable? How can the study help recommend restoration of truly resilient populations of 1000 or more spawners in these watersheds?

Lines 349- Conclusions:

Do you consider the 100 spawners (down from 300) returning to the Nivelle River as NOT ALARMING? You should be less timid in drawing inferences from your study in terms of management. I would consider all of these populations to be exhibiting rather alarming trends.

Referee: 2

Comments to the Author(s)

This clear and well written article uses a previously developed and deployed methodology (Choc) to examine the effects of systematic environmental variation (river temperature and discharge) due to primarily anthropogenic impacts upon the migration patterns of smolt and adult Atlantic salmon. The Choc method, in analyzing patterns based on how changes in the environment are affecting individual animal's environmental niches and the consequent impact of the changes upon population parameters, provides a predictor of population trends that is rooted in ecological (niche) theory. This offers a more satisfying and mechanistic approach to understanding the species' population dynamics compared to the more frequently used correlative approaches. Most of my comments are editorial in nature.

Regarding the title, It seems to me it is not the migration that is in jeopardy, but individual survival during the course of migration and it is survival probabilities that the authors have investigated. I suggest slightly changing the title to capture this, perhaps "Can global change jeopardize Atlantic salmon survival during migrations through joint modifications of river temperature and discharge?"

l. 25 add "fish" before "migration"

l. 33 add "a" before "few"

l. 35 strike the first "then" then add "both life stages" before "preferential". Does that capture what the authors are trying to state? Also, add "these" before "population's".

l. 36 change to "... changes in their local environments due to global change."

l. 54 change "associations" to "combinations" and "subsequently" to "consequently"?

Associations works for the rest of the MS

l. 56 add "The" before "Atlantic salmon"

l. 60 add "typically" before "1-4 years"

l. 62 change "is" to "are"

l. 65 add "minimum" before "temperatures"

l. 70 change "The" to "This"

l. 82 change "population's" to "population"

l. 83 change "their" to "the species' "

l. 88 change "this affinity of the species to the available" to "selected"

l. 101 change to "niches among several rivers. Here it is applied....."

l. 106 change "use" to "used"

l. 145 Were the counting facilities enumerating all fish coming up the river? Or did they only cover part of the river and hence generating a partial count which may not scale among years predictably to the total population size?

l. 153 How did you factor this variable stocking effort into your analysis and among-river comparisons? The numbers are small and perhaps were not expected to appreciably affect either smolt output or adult returns? Please explain.

l. 161 change "salmons" to "salmon"

l. 193 to read " there is a mix of fish that have spent one versus several winters at sea...."

l. 213 change "salmons" to "salmon"

l. 229 add "relatively" before "stable".

l. 230 So the Oir River that had no stocking showed a significant increase in smolt production? What is your interpretation of this?

l. 236 change "expect" to "except for"

l. 284 change "original" to "unique"

l. 290 Can you make this general statement of “profound effects” in the context of the work that you present? You have shown a slow increasing temperature trend and no general patterns for discharge. Are these profound changes?

l. 299 change “between” to “among”

l. 300 change “trophic availability” to “feeding conditions”

l. 303 change to read “drive the motivation to migrate; for example Teichert et al...”

l. 305 to read “migration and a prolongation of the migration period due to climate change....”

l. 308 to read “A longer spring migration season such as that of the Bresle River.....”

l. 314 add “potentially” before “inhospitable”

l. 322 change “salmons” to “salmon”

l. 323 strike “at the moment”

l. 327 add “the” before “spawning”

l. 328 – 348 I do not follow the logic of the statement that the evolutionary potential depends large on the duration of the migration season. Surely it depends on the genetics of the population? This needs to be better explained and defended. The argument surrounding the better capability of a population to adapt if the population has a longer migration period works depending on the definition that is being used for a salmon “population”. If particular tributaries of the system have animals that migrate at different genetically determined times, and conditions deteriorate in the river such that particular runs that occurred at times that are now unfavorable disappear and those occurring in more favorable times surge, is this actually a “population” adaptation as opposed to a replacement of populations? Perhaps the salmon populations in your river are not structured into distinct population segments?

l. 336 change “river” to “rivers”

Conclusion and l. 358 in particular. According to Table 1, the Nivelles River is by far the least disturbed system in which you worked, having 58.6% of its watershed in “natural areas” compared to 3.5% and 19.1% for the Oir and Bresle Rivers, respectively, yet the Nivelles has the highest annual temperatures which seems curious to me. Have these percentages changed over the duration of the study period, or has there been a loss of natural habitat during this time? Since stocking of hatchery fish and habitat restoration are probably the two major activities currently employed for Atlantic salmon restoration, can your methodology and results inform either of these practices?

Table 1, last entry on “Spawners migrating upstream”, did the years included in the total number for each river system vary among systems?

Fig 1 Bresle River panel. To avoid confusion please individually label the Bresle and Yères Rivers in this figure.

Fig 2 and 4 caption. Are the values presented catches? Or are they the counts in the various counting systems used on the different rivers? I would rephrase the description of the axes on both Figures to read that the X and Y axes plot your standardized scores, and the italicized secondary axes plot true values of the variables. The term “natural” is confusing.

Fig 4 caption. Specify that this Figure is for adult salmon.

Author's Response to Decision Letter for (RSPB-2021-1882.R0)

See Appendix A.

Decision letter (RSPB-2021-1882.R1)

12-Nov-2021

Dear Dr Arevalo

I am pleased to inform you that your Review manuscript RSPB-2021-1882.R1 entitled "Does global change increase the risk of maladaptation of Atlantic salmon migration through joint modifications of river temperature and discharge?" has been accepted for publication in Proceedings B.

The referee(s) do not recommend any further changes. Therefore, please proof-read your manuscript carefully and upload your final files for publication. Because the schedule for publication is very tight, it is a condition of publication that you submit the revised version of your manuscript within 7 days. If you do not think you will be able to meet this date please let me know immediately.

To upload your manuscript, log into <http://mc.manuscriptcentral.com/prsb> and enter your Author Centre, where you will find your manuscript title listed under "Manuscripts with Decisions." Under "Actions," click on "Create a Revision." Your manuscript number has been appended to denote a revision.

You will be unable to make your revisions on the originally submitted version of the manuscript. Instead, upload a new version through your Author Centre.

1) A text file of the manuscript (doc, txt, rtf or tex), including the references, tables (including captions) and figure captions. Please remove any tracked changes from the text before submission. PDF files are not an accepted format for the "Main Document".

2) A separate electronic file of each figure (tiff, EPS or print-quality PDF preferred). The format should be produced directly from original creation package, or original software format. Please note that PowerPoint files are not accepted.

3) Electronic supplementary material: this should be contained in a separate file from the main text and the file name should contain the author's name and journal name, e.g. `authorname_procb_ESM_figures.pdf`

All supplementary materials accompanying an accepted article will be treated as in their final form. They will be published alongside the paper on the journal website and posted on the online figshare repository. Files on figshare will be made available approximately one week before the accompanying article so that the supplementary material can be attributed a unique DOI. Please see: <https://royalsociety.org/journals/authors/author-guidelines/>

4) Data-Sharing and data citation

It is a condition of publication that data supporting your paper are made available. Data should be made available either in the electronic supplementary material or through an appropriate repository. Details of how to access data should be included in your paper. Please see <https://royalsociety.org/journals/ethics-policies/data-sharing-mining/> for more details.

<http://datadryad.org/submit?journalID=RSPB&manu=RSPB-2021-1882.R1> which will take you to your unique entry in the Dryad repository.

Once again, thank you for submitting your manuscript to Proceedings B and I look forward to receiving your final version. If you have any questions at all, please do not hesitate to get in touch.

Sincerely,
Dr Daniel Costa
Editor, Proceedings B
mailto:proceedingsb@royalsociety.org

Associate Editor
Comments to Author:

The authors appear to have addressed the two reviewers comments in their revision. I have suggested a small number of editorial clarification which will assist non-experts reading this manuscript in understanding the study and its results easily. If the authors could please make the suggested amendments it would be appreciated.

Reviewer(s)' Comments to Author:

Author's Response to Decision Letter for (RSPB-2021-1882.R1)

See Appendix B.

Decision letter (RSPB-2021-1882.R2)

17-Nov-2021

Dear Dr Arevalo

I am pleased to inform you that your manuscript entitled "Does global change increase the risk of maladaptation of Atlantic salmon migration through joint modifications of river temperature and discharge?" has been accepted for publication in Proceedings B.

Data Accessibility section

Open Access

Paper charges

Sincerely,

Proceedings B

Appendix A

Dear Associate Editor,

We would like to thank the reviewers for their in-depth reading of our manuscript and their useful comments. Please find below the point-by-point answers (in red) to questions of the reviewers and the changes made to our article entitled “*Does global change increase the risk of maladaptation of Atlantic salmon migration through joint modifications of river temperature and discharge?*”. We attached the revised manuscript with the changes highlighted. As the length of the manuscript is limited, we modified the bibliographic list to be able to respond to the reviewers' suggestions as well as possible, with a balanced number of references. Therefore, we addressed the major issue of the manuscript (the lack of a population history) by adding a section in Supplementary Material as this information, although useful, does not change the perspective of our results.

Referee: 1

Comments to the Author(s)

General comments:

This is a good study describing a method to assess the effects of anthropogenic change on aquatic organisms (Atlantic salmon). It should be of broad general interest to conservation-minded biologists.

In order to fully convey the relevance of the analyses undertaken, the authors need to be more explicit about the case studies in both the Methods and the Discussion:

A brief history and summary of population trends must be provided for each study river. Of particular relevance is 1) what were the historic population sizes and distributions within each watershed, and 2) a discussion of how the newly introduced populations (whether from other rivers or from hatcheries) likely differ from the original endemic populations in locally-adapted traits, particularly run timing and age at maturity.

Authors: The monitoring of these populations began in the 1980s with the construction of fish passes and counting stations. Prior to this, it is quite difficult to obtain data on the size of the populations, and consequently very few precise estimations of historic population sizes or distributions are available. Furthermore, stocking operations were implemented before the start of these monitoring programmes, so we cannot currently measure the impact of these operations on the population dynamics. However, stocking operations were carried out using eggs from Scottish and Polish spawners and it is now known that the mortality rates for stocking at this stage with foreign strains are very high. We can therefore assume that the impact of stocking on these populations was limited. Insofar as this information is necessary for the understanding of these three populations but does not influence the results, we added a new part containing all relevant historical information existing in the literature on the history of these populations and the management measures that have been applied in Supplementary Material, Appendix A.

Such small populations as found in the present study (100 adult spawners, probably below minimum viable population size) are likely far below historic levels when locally-adapted sub-populations were probably found throughout each watershed. Each of these subpopulations would have had unique migration adaptations (some of which would have led to a “portfolio effect” a la Schindler et al. 2010, against climate change).

So, while the Choc method may be suitable for understanding trends for the current populations, the authors should be explicit that historic populations would likely have been far more diverse and

resilient to climate change (or other anthropogenic factors).

Authors: The earliest articles mentioning these populations refer to smaller population sizes than those observed today (for example, 10-30 spawners / year on the Nivelle in the 1970s; Dumas & Prouzet 2003; Baglinière 1990). These extremely low population sizes were due to a deterioration of the habitat and especially to the construction of impassable dams in the early 20th century. Despite the small size of these populations, they were maintained and the number of spawners increased significantly when measures to improve habitat quality (reduction of pollution) were taken or when fish ladders were built to allow access to the most productive areas. We added these ideas to better describe the situation of each of these populations in the part dealing with the historic of the populations in Appendix A. In addition, climate change and anthropogenic pressures most likely induced a selection of certain strategies and life history traits but unfortunately our data do not allow us to infer this kind of mechanism.

A few specific comments:

Line 33 "a" few weeks

Authors: As proposed, we modified the sentence (line 33).

Lines 41-43: add "in addition to photoperiod" and appropriate citations

Authors: We revised the sentence to include photoperiod as a key environmental trigger for migration, with appropriate reference in the context of Atlantic salmon (Teichert et al. 2020; lines 42-43). We also underlined the significance of the photoperiod in the introduction (line 66) as well as in the discussion (lines 302).

Lines 48-49: not "emergence" – "evolution" is the correct word

Authors: We corrected the terminology (line 49).

Lines 50-52: "Global change (including climate change and other anthropogenic pressures such as water withdrawal for human consumption, irrigation, industry and hydropower production)". Global change does not include anything that is not global. These other are regional or local!

Authors: We restructured the sentence to emphasise the interaction between global climate change and local anthropogenic pressures and the effects of this interaction on aquatic organisms (lines 51-53).

Lines 64-73: migration timing is a heritable trait, mediated by biotic (internal state) and abiotic (env) factors. Local adaptation in migration timing should be clearly stated here (since it is these local adaptations that are subject to mismatch due to climate change). Again, the historic conditions in the case study watersheds will dictate to some extent the relevance of the analyses.

Authors: Global changes alter environmental conditions and the occurrence of favourable water temperature and discharge simultaneously. When confronted with these new conditions, species could change the timing of their migration to track the same "favourable" conditions (phenological shift), or try to adapt to the new conditions (local adaptation). We specified this explanation in lines 54-57. Regarding the background of these three populations, this is a point that was raised as a major comment and we provided the information to better understand the challenges faced by these populations (Supplementary Material, Appendix A).

Lines 84- why not use the widely accepted terms functional and realized niche?

Authors: The concept of ecological niches is commonly used in ecology. Hutchinson recognizes a species' fundamental niche, a multidimensional 'cloud' of favourable conditions determined by all environmental (abiotic and biotic) variables where the species can reproduce and survive. The functional niche refers rather to the functional role of a species or its position in a food web. These

definitions do not reflect what we were analysing (*i.e.*, the trends within all the environmental conditions offered by the environment). We thought it wise to introduce new terms, “the available niche” and “the preferential niche”, as we had not found a satisfactory equivalent.

Lines 312-317; “In both cases, the smaller length and poorer physical condition of smolts reaching the sea could impair their ability to swim and escape predators, therefore impacting their survival rate [73] and the resilience of smolts to inhospitable marine conditions [74]. Moreover, global changes could lead to a growing mismatch between downstream migration triggers and suitable marine environmental conditions, with potential implications for adult return rates [75] and individual fitness [76,77].”

Is there any evidence compare the historic conditions of smolt size and status, and migration timing mismatch, to the present, highly-modified and small populations? That is, smolt size and timing of these new, small populations might differ substantially from historic populations, that have evolved over thousands of years.

Authors: In the present article, we focused on long-term temporal trends in environmental conditions, examining how conditions historically known as "favourable" for salmon migrations have changes over time and whether they were still available. In response to these environmental changes, we showed in Supplementary material (Figure S2) that the passage date of 95% of smolts on the Bresle River occurs significantly later today than in 1985 and that the passage dates of 5% and 95% of smolts on the Oir River tend to occur earlier in the season. These elements allowed us to discuss a possible mismatch between the migration date of smolts and the favourable conditions at sea (lines 306-314). To support our comments about smolts in “bad conditions” and to follow the recommendations of the reviewer, we added in Supplementary Material the trends in annual averages fork lengths of smolts caught on the Bresle and Oir rivers and we performed Mann-Kendall trend tests on these records. The Mann-Kendall statistics are available on Figure S3. The length of smolts has tended to decrease on the Bresle River ($\tau = -0.23$; p -value = 0.07) and has decreased significantly on the Oir River ($\tau = -0.27$; p -value = 0.03) since 1985. We improved the references to figures and to Mann-Kendall statistics in the text to make it clearer, from line 306 to line 313.

Lines 318- “Diversity of spawning migration strategies and evolutionary potential” :

Again, a discussion of the diversity of the historic, endemic populations is warranted here. How much “shifting baselines” have you experienced. Are populations of 100 spawners even viable? How can the study help recommend restoration of truly resilient populations of 1000 or more spawners in there watersheds?

Authors: Migration strategies were different between these three populations. The spawning migration period was defined between June and November on the Bresle River, between October and December on the Oir River and between April and November on the Nivelles River (Figure 2D and Figure S2). Difference in spawning migration periods among rivers was related to the age structure of the population: on the Oir river, most spawners spent one winter (1SW) at sea (which preferentially migrated in fall), while there was a mix of one winter and several winters at sea (MSW; which migrated preferentially in spring) on the two other rivers (lines 186-192). Consequently, the diversity of migration strategies was higher on the Nivelles and Bresle rivers. However, salmon scale readings are ongoing and the age of salmon caught in traps is not currently available, which prevented us from performing the Choc analysis by year class (lines 319-324).

There is no known critical threshold for Atlantic salmon population below which the number of individuals is too low and the population threatened. For example, the number of spawners on the Nivelles River has significantly reduced since the 2000s (visible in Figure 2 but also observed in Prévost & Lange 2019). However, the number of juveniles (number of parrs observed in fall)

remained stable over time. This might reflect that egg deposition is saturating the currently available habitat. The objective is therefore not to have more than 1000 spawners colonising the watershed but to have a sustainable population. However, if the number of spawners continues to decline, egg deposition by females may no longer saturate the habitat, at which point a decline in parr numbers in fall may be observed. A decline in the number of spawners could reduce the diversity of life history traits, which could reduce the resilience of the population during catastrophic events. We added some elements in this sense in the discussion (lines 346-348).

Lines 349- Conclusions:

Do you consider the 100 spawners (down from 300) returning to the Nivelles River as NOT ALARMING? You should be less timid in drawing inferences from your study in terms of management. I would consider all of these populations to be exhibiting rather alarming trends.

Authors: The size of these populations has been constantly decreasing since the beginning of the monitoring, so it is important to identify the pressures on these populations. Our method revealed mismatches between the preferential conditions and those actually used by smolts on the Bresle River and by spawners on the Oir River. On other stages and/or rivers, environmental changes during migratory events do not seem to explain the decline in these populations. However, there were other pressures that significantly affected the size of these populations (overexploitation, conditions for growth in freshwater or at sea). Consequently, we reformulated the conclusion to emphasize more on the need for management actions to make these populations sustainable (lines 365-366).

Referee: 2

Comments to the Author(s)

This clear and well written article uses a previously developed and deployed methodology (Choc) to examine the effects of systematic environmental variation (river temperature and discharge) due to primarily anthropogenic impacts upon the migration patterns of smolt and adult Atlantic salmon. The Choc method, in analyzing patterns based on how changes in the environment are affecting individual animal's environmental niches and the consequent impact of the changes upon population parameters, provides a predictor of population trends that is rooted in ecological (niche) theory. This offers a more satisfying and mechanistic approach to understanding the species' population dynamics compared to the more frequently used correlative approaches. Most of my comments are editorial in nature.

Regarding the title, It seems to me it is not the migration that is in jeopardy, but individual survival during the course of migration and it is survival probabilities that the authors have investigated. I suggest slightly changing the title to capture this, perhaps "Can global change jeopardize Atlantic salmon survival during migrations through joint modifications of river temperature and discharge?"

Authors: We do not focus directly on fish survival in the Choc analyses but it is true that the potential mismatches revealed by our method can have direct or indirect consequences on the growth, survival and fitness of individuals and their offspring. In response to this comment, we retitled the manuscript as "Does global change increase the risk of maladaptation of Atlantic salmon migration through joint modifications of river temperature and discharge?", maladaptation including threats to the survival and growth of the fish and their offspring.

l. 25 add "fish" before "migration"

Authors: As proposed, we mentioned "fish migration" (line 25).

l. 33 add "a" before "few"

Authors: As proposed, we added "a" before few (line 33).

l. 35 strike the first “then” then add “both life stages” before “preferential”. Does that capture what the authors are trying to state? Also, add “these” before “population’s”.

Authors: As proposed, we edited the sentence (lines 35-37).

l. 36 change to “... changes in their local environments due to global change.”

Authors: We replaced “its” with “their” (line 37).

l. 54 change “associations” to “combinations” and “subsequently” to “consequently”? Associations works for the rest of the MS

Authors: For consistency, we retained the term "association" as we used it throughout the manuscript. (lines 55-56).

l. 56 add “The” before “Atlantic salmon”

Authors: We added “The” (line 58).

l. 60 add “typically” before “1-4 years”

Authors: We did the modification (line 62).

l. 62 change “is” to “are”

Authors: We corrected the sentence (line 64).

l. 65 add “minimum” before “temperatures”

Authors: We made the clarification (line 68).

l. 70 change “The” to “This”

Authors: We replaced “the” with “this” (line 72).

l. 82 change “population’s” to “population”

Authors: We replaced “population’s” with “population” (line 84).

l. 83 change “their” to “the species’ ”

Authors: We followed the suggestion (line 84-85).

l. 88 change “this affinity of the species to the available” to “selected”

Authors: As suggested, we simplified the structure of the sentence (line 90).

l. 101 change to “niches among several rivers. Here it is applied.....”

Authors: We have truncated the sentence as proposed (line 103).

l. 106 change “use” to “used”

Authors: We corrected the sentence (line 108).

l. 145 Were the counting facilities enumerating all fish coming up the river? Or did they only cover part of the river and hence generating a partial count which may not scale among years predictably to the total population size?

Authors: The traps allowed an exhaustive monitoring of salmonid populations (Servanty and Prévost 2016). Even if not all fish would pass through the trap, it is likely that during peak migration periods there are more fish caught by the trap. Transforming our data into daily passage proportions allowed us to better reflect the intensity of migration in relation to environmental conditions without taking into account the population dynamics. We added a clarification and the reference in the text on this subject (line 149-150).

l. 153 How did you factor this variable stocking effort into your analysis and among-river comparisons? The numbers are small and perhaps were not expected to appreciably affect either smolt output or adult returns? Please explain.

Authors: We did not take stocking into account in our analyses. This information is given to characterise each population. Furthermore, given the high mortality rates at the egg stage and the very low quantities of fish restocked, stocking is likely to have had a very limited effect on population dynamics and trends.

l. 161 change “salmons” to “salmon”

Authors: We did the correction (line 159).

l. 193 to read “ there is a mix of fish that have spent one versus several winters at sea....”

Authors: As suggested, we modified the sentence (line 191).

l. 213 change “salmons” to “salmon”

Authors: We did the correction (line 211).

l. 229 add “relatively” before “stable”.

Authors: We added "relatively" (line 227).

l. 230 So the Oir River that had no stocking showed a significant increase in smolt production? What is your interpretation of this?

Authors: The number of spawners remained stable over this period, so the increase in the number of smolts was not linked to an increase in the number of adults. There probably were changes in conditions (environmental conditions, habitat quality, trophic availability) promoting the survival of young stages (eggs, fry, young of the year) which resulted in an increase in the number of smolts produced.

l. 236 change “expect” to “except for”

Authors: We added “for” at line 234.

l. 284 change “original” to “unique”

Authors: We did the modification (line 281).

l. 290 Can you make this general statement of “profound effects” in the context of the work that you present? You have shown a slow increasing temperature trend and no general patterns for discharge. Are these profound changes?

Authors: Looking at the trends revealed within the available niche (for both adults and smolts) as well as the annual trends in Appendix B, there was little change for the Bresle River, the highest discharges were significantly more frequent on the Oir River and the lowest discharges, associated with warm temperatures, were more frequent on the Nivelles River, which represent significant environmental changes for aquatic organisms. We improved the references to the figures to make it easier to understand (lines 285-286).

l. 299 change “between” to “among”

Authors: We modified the manuscript (line 297).

l. 300 change “trophic availability” to “feeding conditions”

Authors: We did the modification (line 298).

l. 303 change to read “drive the motivation to migrate; for example Teichert et al...”

Authors: We modified the sentence (line 301).

l. 305 to read “migration and a prolongation of the migration period due to climate change....”

Authors: As proposed, we changed the wording (line 303).

l. 308 to read “A longer spring migration season such as that of the Bresle River.....”

Authors: We modified the sentence (line 306).

l. 314 add “potentially” before “inhospitable”

Authors: We added “potentially” at line 314.

l. 322 change “salmons” to “salmon”

Authors: We edited “salmons” to “salmon” (line 322).

l. 323 strike “at the moment”

Authors: We deleted “at the moment” (line 323).

l. 327 add “the” before “spawning”

Authors: We added “the” before “spawning” (line 327).

l. 328 – 348 I do not follow the logic of the statement that the evolutionary potential depends large on the duration of the migration season. Surely it depends on the genetics of the population? This needs to be better explained and defended. The argument surrounding the better capability of a population to adapt if the population has a longer migration period works depending on the definition that is being used for a salmon “population”. If particular tributaries of the system have animals that migrate at different genetically determined times, and conditions deteriorate in the river such that particular runs that occurred at times that are now unfavorable disappear and those occurring in more favorable times surge, is this actually a “population” adaptation as opposed to a replacement of populations? Perhaps the salmon populations in your river are not structured into distinct population segments?

Authors: There are generally two migration peaks in salmon populations, a first peak is observed in spring and a second peak in fall. This is the case for the Bresle and Nivelle rivers (Figure 2D). This diversity of strategies (spring migration vs. fall migration) ensures the arrival of spawners on the spawning grounds despite the possible occurrence of a catastrophic event (e.g. flood in spring or prolonged drought in fall). However, on the Oir River, there is only one migration peak in the fall/winter. Therefore, if the spawners cannot access the spawning grounds at time, the whole spawning success is at risk. Furthermore, it has been shown that a diversity of migration strategies is associated with high genetic variability and thus with the ability of the population to adapt to stochastic events and to be more resilient to environmental changes (Sgrò et al. 2011; Kovach et al. 2013, 2015). We added some elements for clarity (lines 330 and 338-340).

l. 336 change “river” to “rivers”

Authors: We did the correction (line 336).

Conclusion and l. 358 in particular. According to Table 1, the Nivelle River is by far the least disturbed system in which you worked, having 58.6% of its watershed in “natural areas” compared to 3.5% and 19.1% for the Oir and Bresle Rivers, respectively, yet the Nivelle has the highest annual temperatures which seems curious to me. Have these percentages changed over the duration of the study period, or has there been a loss of natural habitat during this time?

Authors: The Nivelle River is the southernmost of the rivers considered in this article (almost 1000 km south of the other two rivers; see Figure 1). This difference in latitude explains the warmer water

temperatures throughout the year, in all seasons, observed on the Nivelle River. Moreover, in the predictive models of the IPCC (2014) or also in Dayon et al. (2018), the Nivelle River was included in the Mediterranean region. The percentages of natural areas might have slightly change over time (we don't have precise figures on this aspect), but nothing drastic and clearly not the cause of the temperature differences.

Since stocking of hatchery fish and habitat restoration are probably the two major activities currently employed for Atlantic salmon restoration, can your methodology and results inform either of these practices?

Authors: Our method show whether favourable environmental conditions still match with a key life cycle event. If the environmental conditions change, this method reveals whether the species appears to have been able to adapt to the new conditions. Applied to anthropized rivers, this could reveal a water deficit at a critical period and the management measures that could be taken would be limited water abstraction or water releases (lines 363). In the dataset, the origin of fishes (natural or stocking) was not available; it would have been interested to make separate analysis by origin to explore the influence of restocking though the number of individuals would have probably been too limited.

Table 1, last entry on "Spawners migrating upstream", did the years included in the total number for each river system vary among systems?

Authors: The study period reported in the table (1985-2019, 1987-2019 and 1986-2019 for the Bresle, Oir and Nivelle rivers respectively) covered all data, *i.e.*, water temperature, discharge, smolt and spawner counts. For each river, if a given year was excluded for a given variable (e.g. if data were missing more than 3 consecutive weeks for water temperature or discharge), this year was also excluded for the other variable (e.g. salmon passages). We clarified this at lines 154.

Fig 1 Bresle River panel. To avoid confusion please individually label the Bresle and Yères Rivers in this figure.

Authors: As proposed, we added the Yères River to the figure.

Fig 2 and 4 caption. Are the values presented catches? Or are they the counts in the various counting systems used on the different rivers? I would rephrase the description of the axes on both Figures to read that the X and Y axes plot your standardized scores, and the italicized secondary axes plot true values of the variables. The term "natural" is confusing.

Authors: For Figure 2, we added "caught in the traps" to the caption to make it clear that this referred to the gross number of salmon (388-389). For the Figure 3-4, the values on the Y and X axes represented the empirical cumulative distribution functions of water temperature and discharge. We added natural values in italics to give ecological significance to the quantiles. For example, in figure 4, in the available niche of the Bresle River, 50% of the time the water temperature was below 14.94°C. Similarly, 50% of the time, the discharge was less than 1.85 m³ s⁻¹. We completed the caption accordingly (lines 396-398 and 411-413).

Fig 4 caption. Specify that this Figure is for adult salmon.

Authors: We added "of adults" on line 407.

References cited

Baglinière, J. L., Thibault, M., & Dumas, J. (1990). Réintroductions et soutiens de populations du Saumon atlantique (*Salmo salar* L.) en France. *Revue d'écologie*.

- Dayon, G., Boe, J., Martin, E., & Gailhard, J. (2018). Impacts of climate change on the hydrological cycle over France and associated uncertainties. *Comptes Rendus Geoscience*, 350(4), 141-153.
- Dumas, J., & Prouzet, P. (2003). Variability of demographic parameters and population dynamics of Atlantic salmon *Salmo salar* L. in a south-west French river. *ICES Journal of Marine Science*, 60(2), 356-370.
- IPCC. (2014). Part A: Global and Sectoral Aspects. (Contribution of Working Group II to the Fifth Assessment Report of the Intergovernmental Panel on Climate Change). *Climate Change 2014: Impacts, Adaptation, and Vulnerability.*, 1132. Retrieved from https://www.ipcc.ch/pdf/assessment-report/ar5/wg2/WGIIAR5-FrontMatterA_FINAL.pdf
- Kovach, R. P., Joyce, J. E., Echave, J. D., Lindberg, M. S., & Tallmon, D. A. (2013). Earlier migration timing, decreasing phenotypic variation, and biocomplexity in multiple salmonid species. *PloS one*, 8(1), e53807.
- Kovach, R. P., Ellison, S. C., Pyare, S., & Tallmon, D. A. (2015). Temporal patterns in adult salmon migration timing across southeast Alaska. *Global change biology*, 21(5), 1821-1833.
- Prévost É, Lange F. (2019). *Bilan du suivi du stock de saumon de la Nivelle - Synthèse 1984-2019*. 18 p.
- Servanty S., & Prévost, E. (2016). Mise à jour et standardisation des séries chronologiques d'abondance du saumon atlantique sur les cours d'eau de l'ORE DiaPFC et la Bresle (Doctoral dissertation, auto-saisine).
- Sgrò, C. M., Lowe, A. J., & Hoffmann, A. A. (2011). Building evolutionary resilience for conserving biodiversity under climate change. *Evolutionary applications*, 4(2), 326-337.
- Teichert, N., Benitez, J. P., Dierckx, A., Tétard, S., De Oliveira, E., Trancart, T., ... & Ovidio, M. (2020). Development of an accurate model to predict the phenology of Atlantic salmon smolt spring migration. *Aquatic Conservation: Marine and Freshwater Ecosystems*, 30(8), 1552-1565.

Appendix B

Dear Editor,

We would like to thank the associate editor for the in-depth reading of our manuscript and the useful comments. We return the modified version of the manuscript to which we applied all the changes suggested by the associated editor.